# Intraspecific Variation in Functional Traits of *Medicago sativa* Determine the Effect of Plant Diversity and Nitrogen Addition on Flowering Phenology in a One-Year Common Garden Experiment

**DOI:** 10.3390/plants12101994

**Published:** 2023-05-16

**Authors:** Yue Ma, Xiang Zhao, Xiaona Li, Yanxia Hu, Chao Wang

**Affiliations:** 1College of Grassland Science, Shanxi Agricultural University, Jinzhong 030801, China; 2Institute of Grassland, Flowers and Ecology, Beijing Academy of Agriculture and Forestry Sciences, Beijing 100097, Chinasunny3629@126.com (Y.H.)

**Keywords:** functional traits, abiotic factors, flowering phenology, nitrogen addition, plant diversity, common garden

## Abstract

Nitrogen deposition and biodiversity alter plant flowering phenology through abiotic factors and functional traits. However, few studies have considered their combined effects on flowering phenology. A common garden experiment with two nitrogen addition levels (0 and 6 g N m^−2^ year^−1^) and five species richness levels (1, 2, 4, 6, and 8) was established. We assessed the effects of nitrogen addition and plant species richness on three flowering phenological events of *Medicago sativa* L. via changes in functional traits, soil nutrients, and soil moisture and temperature. The first flowering day was delayed, the last flowering day advanced, and the flowering duration shortened after nitrogen addition. Meanwhile, the last flowering day advanced, and flowering duration shortened along plant species richness gradients, with an average of 0.64 and 0.95 days change per plant species increase, respectively. Importantly, it was observed that plant species richness affected flowering phenology mainly through changes in plant nutrient acquisition traits (i.e., leaf nitrogen and carbon/nitrogen ratio). Our findings illustrate the non-negligible effects of intraspecific variation in functional traits on flowering phenology and highlight the importance of including functional traits in phenological models to improve predictions of plant phenology in response to nitrogen deposition and biodiversity loss.

## 1. Introduction

Human activities such as agricultural production and fossil fuel combustion add ~200 Tg/year of nitrogen (N) to global ecosystems, approximately equal to that provided by natural N fixation [1]. Land use changes, climate changes, and N deposition increase are the major causes of biodiversity loss [2,3]. Numerous studies showed that plant functional traits and ecosystem functions changed significantly under N deposition increase and biodiversity loss [4,5,6]. Phenology, the periodic events in the life cycles of plants, which is sensitive to global changes [7], and thus understanding the influence of N addition, plant diversity, and its interactions on plant phenology is essential.

Flowering phenology, including flowering date and duration, is an important developmental stage in plant phenology and is known to be determined by numerous abiotic factors, such as temperature, photoperiod, and resource availability [8,9]. Recent research revealed that plant functional traits, such as plant height and leaf traits, may serve as direct determinants for flowering phenology [10,11,12]. For example, plant height is critical for a species’ competitive ability to capture light; taller species with higher maximized light interception and faster relative growth rate often flower later than shorter ones [13,14]; specific leaf area (SLA) is linked to growth rates, and higher SLA was found to be associated with stronger phenological shifts [15]; leaf carbon and N content are measures for the plants’ investment in structural components and photosynthetic accumulation [16,17], whereas shifts in flowering phenology could be due to a trade-off between the investment between vegetative and reproductive growth [18]. Although the links between plant functional traits and flowering phenology have been reported, little is known about which traits are better for predicting the changes in flowering phenology.

Functional traits are measurable biotic properties related to the adaptation of plants to the environment [19,20]. Nitrogen addition stimulates plant growth through increasing soil-available nitrogen, resulting in changes in functional traits, such as plant height, leaf area, and nitrogen content, but varied with plant species [21,22]. Multiple changes in the biotic and abiotic environments along plant diversity gradients require a coordinated response in numerous traits to achieve a balance among different functions [4,23]. For instance, plants with slow growth rate may adjust morphological traits to tolerate low light availability through the formation of longer and thinner leaves [24]; variation in leaf nitrogen content (LNC) may serve as a physiological adjustment to improve photosynthetic carbon gain [25,26]. Hence, plant flowering phenology was significantly affected by N addition [27,28] and plant diversity [5,29], while our knowledge regarding how plant diversity and N deposition affect flowering phenology through plant functional traits was essential to step forward.

Here, we report on a common garden experiment investigating the influence of N addition and plant species richness on three flowering phenologies of *Medicago sativa* L. in an assemblage grassland (Appendix A). Specifically, we explored the effects of N addition and plant species richness on the first flowering day (FFD), last flowering day (LFD), flowering duration (FD), flower numbers (FN), environmental factors, and a series of functional traits. We hypothesized that: (a) increased N inputs and plant species richness would delay flowering phenology; (b) increased N inputs promote light acquisition traits (e.g., SLA and plant height), while decreased plant species richness promotes nutrient acquisition traits (e.g., leaf N concentration); and (c) intraspecific variation in functional traits and changes in abiotic factors would co-drive the response of flowering phenology to N addition and plant species richness.

## 2. Results

### 2.1. The Effects of N Addition and Plant Diversity on Soil Environmental Conditions

In our common garden experiment, soil temperature and soil moisture at a depth of 10 cm were significantly influenced by nitrogen addition and plant species richness (Table 1). Soil temperature significantly increased, but soil moisture did not change after nitrogen addition (Table 1 and Figure 1). Soil temperature and soil moisture changed markedly following plant species richness increase (Figure 1c,d). Soil temperature significantly decreased along plant species richness gradients under nitrogen addition (Figure 1c). Soil moisture showed no significant relationship with plant species richness under control and nitrogen addition treatments (Figure 1d).

### 2.2. The Effects of N Addition and Plant Diversity on Flowering Phenology

The flowering phenology of *M. sativa* was significantly affected by nitrogen addition and plant species richness (Table 1), but the effects varied in different phenological phases (Figure 2). Nitrogen addition delayed the first flowering day of *M. sativa* by 1.9 ± 0.39 days, but plant species richness had no effects on it (Figure 2a). The last flowering day advanced by 1.2 ± 0.21 days after nitrogen addition (Figure 2b) and advanced by an average of 0.51 and 0.77 days per plant species increase under control and nitrogen addition treatments, respectively (Figure 2b). The delay of the first flowering day and the advancement of the last flowering day inevitably led to the shortening of the flowering duration. Nitrogen addition shortened the flowering duration of *M. sativa* by 3.2 ± 0.31 days, and it was shortened by 1.08 and 0.82 days per species increase under control and nitrogen addition treatments, respectively (Figure 2c). Moreover, our study also showed that flower numbers significantly decreased after nitrogen addition but increased along the plant species richness gradients (Figure 2d).

### 2.3. The Effects of N Addition and Plant Diversity on Functional Traits

Nine different leaf traits of *M. sativa* were measured in our study. For light acquisition traits, nitrogen addition significantly increased leaf area but did not change leaf mass and specific leaf area (Table 1 and Figure 3). Specific leaf area significantly decreased with increasing plant species richness (Figure 3), but not for other traits (Table 1 and Figure 3). For nitrogen acquisition traits, nitrogen addition had minor effects on leaf carbon content, leaf nitrogen content, and leaf carbon/nitrogen ratio (Table 1). Leaf nitrogen content significantly decreased, but the leaf carbon/nitrogen ratio significantly increased with increasing plant species richness (Figure 3d,e). There is no interaction effect between nitrogen addition and plant species richness on functional traits (Table 1).

### 2.4. Ecological Factors Influencing Flowering Phenology

The last flowering day was positively correlated with the first flowering day significantly (Figure 4a); flowering duration was negatively correlated with the first flowering day and positively correlated with the last flowering day (Figure 4b,d); flower numbers were positively correlated with last flowering day and flowering duration (Figure 4e,f).

Variation partitioning analysis indicated that nutrient acquisition traits and abiotic factors explained a much greater portion of the variance in the first flowering day (31% and 15%, Appendix A), last flowering day (23% and 44%, Appendix A), flowering duration (33% and 46%, Appendix A), and flower numbers (32% and 44%, Appendix A), respectively. The structural equation models showed that the changes in the first flowering day were positively correlated with soil-available nitrogen but negatively correlated with leaf nitrogen content and leaf carbon/nitrogen ratio (Figure 5a). The negative effects of plant species richness on the last flowering day and flowering duration were mainly through its negative effects on leaf nitrogen content (Figure 5b,c). Plant species richness impacted flower numbers directly or through its negative effects on leaf carbon/nitrogen ratio indirectly (Figure 5d), and nitrogen addition impacted flower numbers directly or through its positive effects on leaf mass indirectly (Figure 5d).

## 3. Discussion

### 3.1. The Effects of N Addition and Plant Diversity on Flowering Phenology

The flowering phenology is one of the more important factors determining the reproductive success of plants because the timing of flowering represents when plants expose their reproductive organs to the changing biotic and abiotic environments [30,31]. Flowering phenology changed after N addition in our study was consistent with the results from natural grasslands [27,32]. Soil-available N increased following N addition, which delayed the plant switches from vegetative to reproductive growth and shortened the reproductive duration [28], resulting in a delay in FFD and advancement in LFD. FD was positively correlated with LFD and negatively correlated with FFD, and then the FD was shortened after N addition. A previous study found that plants delayed the switches from vegetative to reproductive growth due to the stimulation of growth through enhancement in soil N availability [28], which led to lower investment in plant reproduction, resulting in a reduction in FD in the N addition treatment.

Similar to the effects of N addition, plant species richness did not change FFD but advanced LFD, which resulted in a shortening in FD. The effects of plant species richness on FFD was inconsistent with the results (ranging from advancement of 1.8 d per species lost to a delay of 0.7 d per species lost; the average is 0.6 d advance) reported from the serpentine grasslands in North America [5], suggesting that the effects of plant species richness on FFD varied among plant species. Moreover, we noticed that the amplitude of changes (0.64 d) in LFD was larger than that in FFD after per species change, which revealed that LFD might be more sensitive to plant species richness. Soil-available N increased along the plant species richness gradients (Appendix A), which delayed the plant switches from vegetative to reproductive growth and shortened reproductive duration [28], resulting in advancement in LFD. FD was positively correlated with LFD, resulting in a shortening in FD.

### 3.2. The Effects of N Addition and Plant Diversity on Functional Traits

We found evidence to support our second hypothesis that increased N inputs promote light acquisition traits. Specifically, the leaf area of *M. sativa* was promoted after N addition may vary due to the increasing demand for light [33]. These results were inconsistent with that reported on the grasses and forbs from the alpine grasslands in China [6,34], which may be partly caused by the differences in plant species. *M. sativa*, legumes, can fix N using symbionts to relieve their N limitation [35], and thus N addition had no effects on the nutrient acquisition traits of *M. sativa*, which may lead to no interactions with plant species richness.

Our findings for plant species richness also support our second hypothesis that both light (SLA) and nutrient (LNC and relative abundance) acquisition traits were promoted with decreasing plant species richness. These results were inconsistent with the results reported from the grasslands in Germany [4], which may be caused by the differences in plant species. Closed canopies are characterized by pronounced gradients in spectral light quality and quantity [36]. In the present study, plant species richness impacts on morphological traits associated with light acquisition were mainly attributable to the changes in the relative height of *M. sativa*. Relative height increased along the plant species richness gradients, indicating less effort for light acquisition should be allocated by plants [37,38]. Hence, *M. sativa* exhibited low SLA in adjustment to decreasing competition for light. Moreover, nitrogen is not the limited resource of *M. sativa*. Lower light acquisition traits are accompanied by lower nutrient acquisition traits [33], resulting in a decrease in LNC along with plant species richness gradients. 

### 3.3. Intraspecific Variation in Functional Traits Determines Flowering Phenology

Our findings did not support our third hypothesis that intraspecific variation in functional traits and alteration in abiotic factors co-drove the response of flowering phenology to N addition and plant species richness; the contributions of intraspecific variation in functional traits for changes in flowering phenology was far greater than abiotic factors (Figure 5 and Appendix A). Specifically, the delay in FFD along the plant species richness gradients was mainly induced by an increase in soil-available N, which was consistent with the result from Wolf et al. in American serpentine grasslands [5]. Higher soil available N delayed the plant switches from vegetative to reproductive growth [28] and then led to the delay in FFD in the monocultures. However, the changes in LFD and FD were mainly driven by the intraspecific variation in LNC. Leaf nutrients could influence the strength of shifts in flowering phenology as plant performance is positively associated with phenological shifts [15,39] and the alteration in the natural selection of plants [12,40,41]. In the monocultures, lower intraspecific competition for resources and light [38] led to higher LNC and SLA and minimized light interception [42], resulting in advancing FFD, delaying LFD, and extending FD (Figure 2). As plant species richness increases, the intraspecific competition decreases, but the interspecific competition among species increases [43,44]. *M. sativa* is more conservative in resource utilization, with low SLA, LNC, and maximized light interception, and thus the “slow” growth strategy [45,46,47] led to advancement in LFD and shortened in FD. Therefore, the intraspecific variation in leaf traits caused the alteration in the growth strategy of *M. sativa* due to interspecific competition increased with increasing plant species richness, which resulted in changes in flowering phenology. Furthermore, the survival pressure of *M. sativa* increased due to higher interspecific competition in the mixed communities, which may cause *M. sativa* to invest more resources in reproductive growth [10], resulting in a positive relationship between FN and plant species richness.

## 4. Materials and Methods 

### 4.1. Study Site and Experimental Design

Our study site (40°10′ 45″ N, 116°26′ 13″ E, and 50 m above sea level) is located at the Station of the Institute of Grassland, Flowers and Ecology on Xiaotang Mountain in Beijing, China. The mean annual precipitation is 526 mm, the mean annual temperature is 11.8 °C (2000–2018) in our study site. The common garden experiment was established in 2019, using a split-plot experiment design with N addition (0 and 6 g N m^−2^ year^−1^) as the main treatment factors and plant species richness (species richness is 1, 2, 4, 6, and 8) as the subfactors (Appendix A). The experiment comprised the following: 2 N addition levels × 4 replicate blocks × 17 combinations of species per N treatment/replicate (four combinations per level of species richness with 1, 2, 4, and 6 species and one with 8 species) = 136 pipes. Each pipe (30 cm in diameter and 50 cm in height polyvinylchloride sealed pipes) was filled with uniformly mixed soil (natural soil with stones and roots sieved out) with sand (in a 3:1 soil–sand ratio) and then buried into the ground without any shelters. Eight plant species were selected in our study to build assembled grassland communities [48]. Different species of seeds were mixed up well and distributed randomly in the pipes, maintained at around 60 individuals in each pipe at the beginning of the experiment, and did not re-sow later. Nitrogen was added as urea (12.86 g m^−2^ year^−1^), dissolved in N-free water, and then applied by spraying on 1 May of each year; the control treatment received equal N-free water. Additional information regarding the experimental design was provided by Wang et al. [48].

### 4.2. Phenology Monitoring 

To track the flowering phenology of *M. sativa*, phenology was monitored every 3–4 days during the growing season from May to September 2019. Three individuals in each pipe were randomly selected, marked, and monitored across the growing season. The first and last date of a flower observed for each marked individual was recorded as the first and last flowering day, and the periods between the first and last flowering day were recorded as flowering duration. Flowering numbers were counted for each marked individual. Flowering phenological events and flowering numbers were averaged for three individuals in each pipe.

### 4.3. Functional Traits and Abiotic Factors Measurements

Light acquisition traits (plant height and relative height, leaf mass and area, leaf length and width, and specific leaf area) and nutrient acquisition traits (leaf carbon content, leaf nitrogen content, leaf C/N ratio, abundance and relative abundance of plant species) are closely related to plant phenology [49,50]. Consequently, we determined these traits to explore the mechanism underlying regulating the response of flowering phenology to experimental N addition and plant species richness in the peak growing season in 2019. Before the measurements, we investigated the abundance and height of each plant species in the pipes. *M. sativa* is the predominant species (relative abundance >40% in each pipe) (Appendix A), three healthy individuals, which we measure flowering phenology were selected to measure the species-level traits in each pipe, and six leaves on each individual were selected to measure leaf traits once at full flowering, and the leaves in each pipe were collected on the same day. The traits were quantified using standard methods proposed by Pérez-Harguindeguy et al. [51]. Specific leaf area was calculated as the ratio of leaf area to its dry weight. Leaf area, length, width, and maximum width, spread leaves were scanned and analyzed by Li-Cor 310 (Li-Cor Inc., Lincoln, USA), and then leaves were oven-dried to a constant weight. The oven-dried leaf samples were ground to determine leaf C and N concentration with an elemental analyzer (PE 2400 II, PerkinElmer Ltd., CT, USA) and then to calculate the C/N ratio. To measure the biomass of *M. sativa*, the aboveground part of each pipe was clipped in early September (the peak of the growing season) in 2019. *M. sativa* clipped from each pipe were pooled together and then oven-dried at 65 °C to a constant weight. 

Soil temperature and moisture at a depth of 10 cm were measured every week from April to October with a W. E. T sensor kit (Delta-T Devices Ltd, Cambridge, UK). Three soil cores were collected in each pipe in early September at a depth of 10 cm and then mixed into one sample. Available-soil N (Ammonium (NH_4_^+^) and nitrate (NO_3_^−^)) concentrations in the extracts were determined calorimetrically by automated segmented flow analysis (Bran + Luebbe AAIII, Bran + Luebbe Ltd, Hamburg, Germany).

### 4.4. Statistical Analyses

We analyzed experimental data with the following three steps. First, we scaled the species-level height to the community level by calculating the mean of abundance distributions (Equation (1) [52]):
(1)$$Mean_{c}=\sum_{i}^{n}p_{i}T_{i}$$
where $p_{i}$ and $T_{i}$ are the relative abundances and the plant height of the species *i*, respectively, and *n* is the number of plant species. Hence, the average height of *M. sativa* divided by *Mean_c_* is the relative height.

Second, we applied linear mixed-effects models using the “*lme*” function (package “*nlme*” [53]) to test the effects of N addition and plant species richness on soil temperature and moisture. We set N addition and species richness levels as fixed effects; the date, block, and plant combination were set as random effects in each model to account for variation among repeated measurements. In addition, linear mixed-effects models were also used to examine the effect of N addition and plant species richness on flowering phenology and functional traits. Nitrogen addition and species richness levels were treated as fixed effects, and the block and plant combinations were treated as random effects. 

Third, variation partitioning analysis that partitioned the variance shared by all factors was then used to quantify the unique contribution of biotic and abiotic factors. Structural equation modeling was employed to evaluate which are the major factors that influence flowering phenology [54] by the package ‘*piecewise-SEM*’ in R software [55]. The SEM requires establishing an a priori framework based on the hypothesized causal relationships among these variables. Second, the relationships between these variables were examined by bivariate correlations. Finally, models with lower *Fisher’s C* and Akaike information criterion (*AIC*) and higher *p* values (*p* ≥ 0.05) were selected in our analysis (Figure 5). All statistical analyses and graphs were prepared in R 3.2.2 [56]. Differences were considered to be statistically significant at *p* ≤ 0.05.

## 5. Conclusions

The study highlighted the influence of functional traits on flowering phenology following nitrogen addition levels and plant species richness gradients in an assemblage grassland through a common garden experiment. It was observed that the first flowering day was delayed 0.31 days, the last flowering day advanced 0.64 days, and the flowering duration was shortened by 0.95 days with per-plant species increase, but the effects of plant species richness on flowering phenology did not interact with nitrogen addition, which indicates that nitrogen addition could change plant flowering phenology by changing biodiversity, but the effects would be independent with the effects of biodiversity. Moreover, flowering phenology changed following nitrogen addition levels, and plant species richness gradients were mainly driven by the intraspecific variation in functional traits, which suggests that variation in functional traits among communities may be a good predictor for the dynamic of plant phenology under global changes. 

## Figures and Tables

**Figure 1 plants-12-01994-f001:**
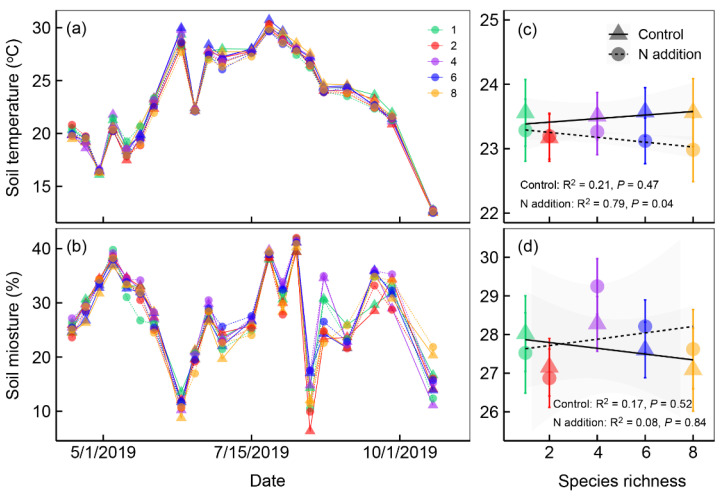
Soil temperature (°C, at a depth of 10 cm) (**a**) and soil moisture (%, at a depth of 10 cm) (**b**) under different plant species richness and nitrogen addition levels from April to October 2019. Mean (±*se*, *n* = 4 or 8) soil temperature (**c**) and soil moisture (**d**) for the whole growing season along the plant species richness gradients under different nitrogen (N) addition levels in 2019. We applied the ordinary linear regression analysis to test the dynamic of soil temperature and soil moisture along plant species richness gradients. Points and lines with different shapes represent different nitrogen addition levels, and points with different colors represent different plant species richness levels.

**Figure 2 plants-12-01994-f002:**
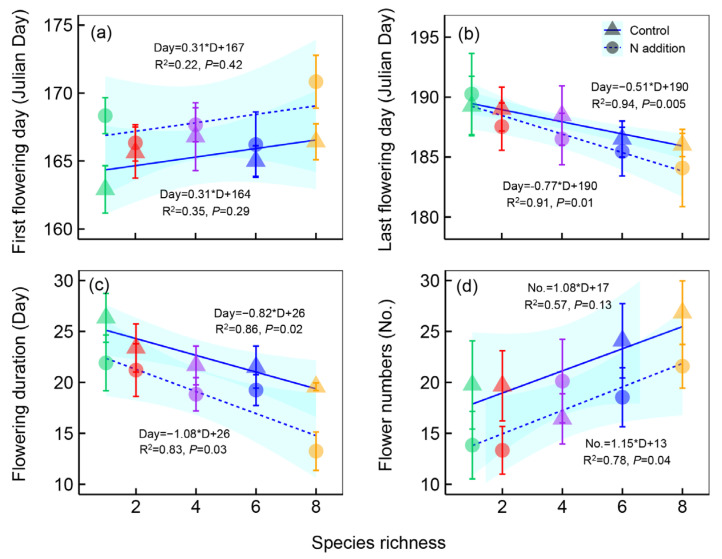
The first flowering day (**a**), last flowering day (**b**), flowering duration (**c**), and flower numbers (**d**) of *Medicago sativa* along the plant species richness gradients under different nitrogen (N) addition levels. All the analyses were performed using the ordinary linear regression analysis to test the dynamic of flower phenological events along plant species richness gradients. Points and lines with different shapes represent different N addition levels, and points with different colors represent different species richness levels. Shaded areas show 95% confidence intervals of the fit.

**Figure 3 plants-12-01994-f003:**
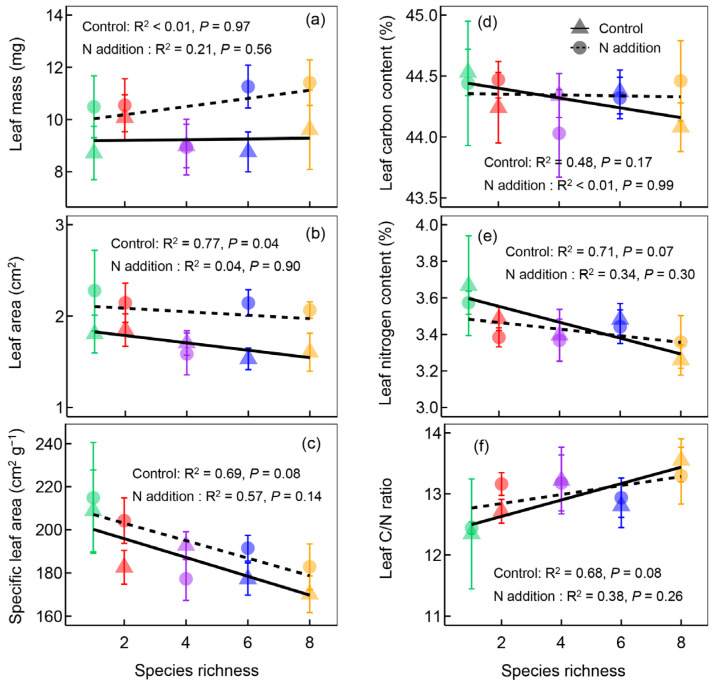
The leaf mass (**a**), leaf area (**b**), specific leaf area (**c**), leaf carbon content (**d**), leaf nitrogen content (**e**), and leaf carbon/nitrogen ratio (**f**) of *Medicago sativa* along the plant species richness gradients under different nitrogen (N) addition levels. Points and lines with different shapes represent different nitrogen addition levels, and points with different colors represent different plant species richness levels.

**Figure 4 plants-12-01994-f004:**
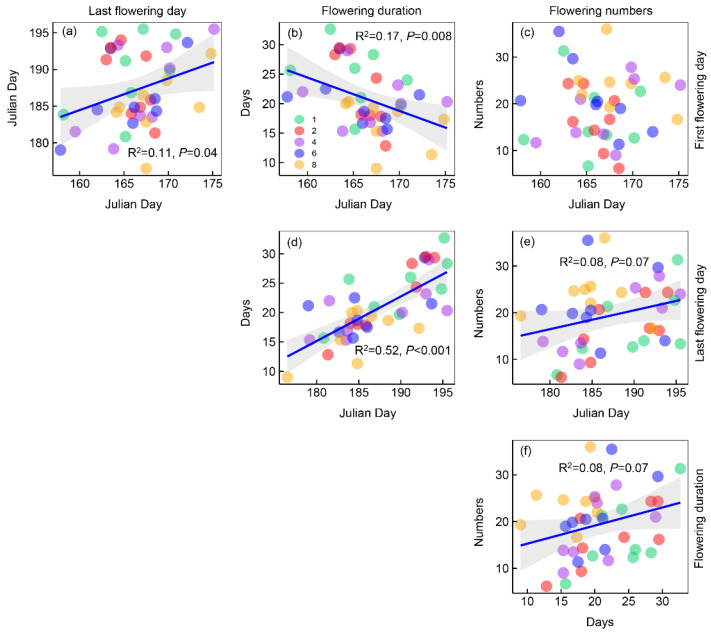
Plots showing the relationships between the four metrics of phenology. Points with different colors represent different plant species richness levels. Shaded areas show 95% confidence intervals of the fit.

**Figure 5 plants-12-01994-f005:**
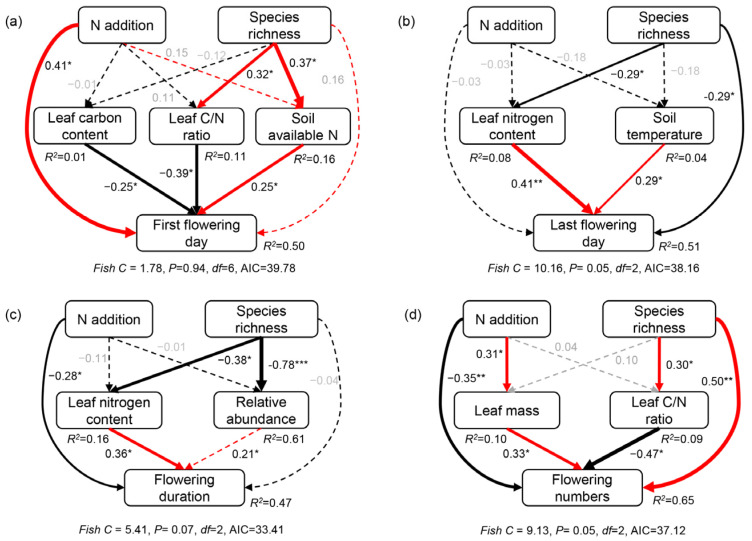
Structural equation modellings of plant species richness and nitrogen (N) addition impact on the first flowering day (**a**), last flowering day (**b**), flowering duration (**c**), and flower number (**d**). Red and black arrows represent significant positive and negative pathways, respectively. Solid and dashed arrows indicate significant and non-significant pathways, respectively. Numbers near the arrow indicate the standardized path coefficients indicating the effect size of the relationship with * indicating *p* < 0.05, ** indicating *p* < 0.01, and *** indicating *p* < 0.001. The arrow width is proportional to the strength of the relationship. *R*^2^ represents the proportion of variance explained for each dependent variable. The goodness-of-fit statistics for the structural equation modeling are shown below each model. C, carbon.

**Table 1 plants-12-01994-t001:** Linear mixed-effects model results showing the effects of plant species richness and nitrogen addition treatments on biotic and abiotic factors.

Parameters	Nitrogen Addition			Species Richness			Nitrogen Addition × Species Richness		
	*numDF*	*F*	*P*	*denDF*	*F*	*P*	*denDF*	*F*	*P*
**Soil moisture**	1	1.99	0.16	4	7.61	**<0.01**	4	1.42	0.23
**Soil temperature**	1	44.62	**<0.01**	4	4.52	**<0.01**	4	6.33	**<0.01**
**Soil available nitrogen**	1	0.15	0.71	4	5.94	**0.01**	4	0.36	0.55
**Leaf mass**	1	4.07	0.06	4	0.17	0.68	4	0.71	0.41
**Leaf area**	1	7.15	**<0.01**	4	1.43	0.24	4	0.41	0.53
**Specific leaf area**	1	1.61	0.21	4	8.51	**<0.01**	4	0.02	0.97
**Leaf carbon content**	1	0.01	0.97	4	0.26	0.61	4	0.05	0.82
**Leaf nitrogen content**	1	0.41	0.53	4	4.15	**<0.01**	4	0.41	0.53
**Leaf carbon/nitrogen ratio**	1	0.25	0.62	4	3.91	**<0.01**	4	0.24	0.63
**First flowering day**	1	2.52	0.09	4	2.01	0.15	4	0.01	0.91
**Last flowering day**	1	2.23	0.09	4	20.05	**<0.01**	4	0.46	0.49
**Flowering duration**	1	8.62	**<0.01**	4	13.09	**<0.01**	4	0.23	0.64
**Flowering numbers**	1	6.38	**0.01**	4	14.31	**<0.01**	4	0.02	0.89

## Data Availability

The data that support the findings of this study are available from the corresponding author upon reasonable request.

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
