# Peer review of "Intraspecific Variation in Functional Traits of Medicago sativa Determine the Effect of Plant Diversity and Nitrogen Addition on Flowering Phenology in a One-Year Common Garden Experiment"

_plants, 2023, doi:10.3390/plants12101994_

Round 1

Reviewer 1 Report

Authors presented a garden experiment with two N addition and six of Medicago sativa diversity levels: three flowering phenology events (first flowering day (FFD) delayed, last flowering day (LFD) advanced, and flowering duration (FD) shortened after N addition).

It is very well-designed study, presented in a manuscript with adequate graphics and statistics and discussion and conclusion sections based on the results.

References are ok and at the journal format, but could be updated to include more papers from the last 4 years.

Suplemmentary file is ok.

Highlights and graphical abstract were not presented.

At line 214, please observe the citation number.

Please, normalize et al. in italics.

Please, observe the plant name with species in low caps.

Reviewer 2 Report

Interesting project and a time and money-consuming one. In my opinion, there are several topics to improve this manuscript. First, the experimental design is not completely clear enough, you should say precisely what is the main plot and the subplots; second, how many times did you measure the variables? (The time factor or repeated measurement factor). Figure 1 of the supplementary files shows the design where is not clear if the treatments or levels factors were randomized. 

Table 1 of the manuscript includes the degree of freedom of each factor; however, the time factor was not included as well as their interactions with the other factors; on the other hand, every factor and the interaction in table one has the same degree of freedom which is not possible in a split-plot design. You have two levels of N and 17 combinations of species. How do end up with 5 levels for the second factor?  That is not explained in the M&M chapter. What is the difference between the colors black, violet, red, and green? I suppose the graphs of the means are good, but the significant differences may change if you use the correct model to analyze the data.

I suggest for phenology the use of degree days instead of calendar days. The duration of any stage is not fixed for days, it is for degree days. 

Reviewer 3 Report

In material and methods analysis are not adequately written. The research contucted was performed on a small set of observed properties.

Reviewer 4 Report

This is an important and timely contribution to plant science.

I recommend publication after certain amendments, as follows:

Abstract: lines 17-20: the description is a bit vague: explicitly mention which abiotic factors and which diversity levels.

lines 20-23: as the changes seem short, were these statistically significant?

Results: FFD does not occur earlier, the result is not statistically significant according to Fig. 2a, so please rephrase everywhere and do not emphasise so much on it.

Figs. 3a and 3b do not have trend lines, why? Please add.

Fig. 5 and generally regarding your statistical approach: you have collinear variables and therefore you expect overfitting in your model. How did you deal with this? If you either penalise inter-dependence among your independent variables, or if you run a test beforehand, you might kick out some of these variables. Plese check this and justify and discuss.

Discussion, section 3.1: FFD does not significantly advance, please avoid emphasising so much on it!

lines 207-212: be careful because here you compare with an alpine grassland, with other environmental stressors as well. Also, the diversity of course up there would be much lower to the garden studied here.

lines 227-231: re-visit this after you check my comment on the statistical approach.

Overall, you do not refer at all to FN: also, another trait that could have been studied is the pollen numbers per flower or per plant. Please mention this as a limitation of the study.

Methods: lines 275-277: since you define FFD and LFD as the first and last appearance of flowers, would you expect any differences if you had used a different approach? Could you provide (for review purposes only) the flowering season per plot or treatment (since you also measured the FN)? Did you have long tails in the start or end of the flowering?

lines 277-279: how were the 3 individuals selected? The functional traits studied were also coming from these 3 individuals or were randomly taken? Also, all measurements were done every day, every week or how often?

lines 308-314: did you try (or could you have tried) checking everything in a full factorial model instead of pooling or creating baselines? This might give even more information. Please discuss and justify.

Finally, English has several flaws throughout the manuscript, please edit thoroughly.

Round 2

Reviewer 1 Report

The new version of the manuscript has been completely redrafted and certainly improved. In fact, there are still some issues to watch out for.

The Graphic Summary is confusing and should be reworked.

The wording should be revised: "We observed..." should be changed to "It was observed...", from the abstract.

Authors should analyze the feasibility of using so many references. The first two short paragraphs of the introduction feature 40 references, only 20% of which would be most appropriate. At the end of this section, 63 references, about 30% more than the previous version. At the end of the manuscript, 91 references, certainly some of them unnecessary, similar to a review article. Mainly for this reason, the manuscript is indicated to be rejected, as the necessary changes would certainly take several weeks longer than those already carried out in the different versions of the manuscript already submitted.

Reviewer 2 Report

Thanks for the changes made in your manuscript. However, you should include the interaction between date and factor in your model; if you omit them, those sum squares go to enlarge the error term or denominator of the F test and consequently it will be more difficult to detect statistical differences. 

On the other hand, as you know, the phenological changes or development basically depend on the ambient temperature; so report the appearance and duration in terms of days could not adequately results presentation because the climate change and weather variability.

Please consider these recommendations to improve your paper. 

Reviewer 3 Report

Autors made significant changes in the paper according to the instructions of reviewers.

Reviewer 4 Report

Thank you for the efficient response to my comments. The work deserves scientific merit and is worth publishing in the Journal.

Round 3

Reviewer 1 Report

Authors performed all modifications asked by reviewers.

Reviewer 2 Report

Everything looks good.